# Quartz-Enhanced Photoacoustic Sensor Based on a Multi-Laser Source for In-Sequence Detection of NO_2_, SO_2_, and NH_3_

**DOI:** 10.3390/s23219005

**Published:** 2023-11-06

**Authors:** Pietro Patimisco, Nicoletta Ardito, Edoardo De Toma, Dominik Burghart, Vladislav Tigaev, Mikhail A. Belkin, Vincenzo Spagnolo

**Affiliations:** 1PolySense Lab-Dipartimento Interateneo di Fisica, Politecnico and University of Bari, Via Amendola 173, I-70100 Bari, Italy; n.ardito2@studenti.uniba.it; 2Polysense Innovations Srl 2, Via Amendola 173, I-70100 Bari, Italy; 3Walter Schottky Institute, Technical University of Munich, Am Coulombwall 4, D-85748 Garching, Germany; edoardo.detoma@wsi.tum.de (E.D.T.); dominik.burghart@wsi.tum.de (D.B.); vladislav.tigaev@wsi.tum.de (V.T.); mikhail.belkin@wsi.tum.de (M.A.B.)

**Keywords:** quartz-enhanced photoacoustic spectroscopy, air pollutants, quantum cascade lasers

## Abstract

In this work, we report on the implementation of a multi-quantum cascade laser (QCL) module as an innovative light source for quartz-enhanced photoacoustic spectroscopy (QEPAS) sensing. The source is composed of three different QCLs coupled with a dichroitic beam combiner module that provides an overlapping collimated beam output for all three QCLs. The 3λ-QCL QEPAS sensor was tested for detection of NO_2_, SO_2_, and NH_3_ in sequence in a laboratory environment. Sensitivities of 19.99 mV/ppm, 19.39 mV/ppm, and 73.99 mV/ppm were reached for NO_2_, SO_2_, and NH_3_ gas detection, respectively, with ultimate detection limits of 9 ppb, 9.3 ppb, and 2.4 ppb for these three gases, respectively, at an integration time of 100 ms. The detection limits were well below the values of typical natural abundance of NO_2_, SO_2_, and NH_3_ in air.

## 1. Introduction

Air pollution refers to the modification of the characteristics of the atmosphere induced by contaminants [1]. Sources of air pollution are, for example, any kind of combustion, vehicles, and industrial processes that send the waste of production into the atmosphere [2]. Air pollutants can be classified as either primary or secondary air pollutants [3]. Primary air pollutants include constituents emitted directly into the atmosphere from the source. Secondary air pollutants are generated by chemical reactions involving primary pollutants and other atmospheric constituents [4]. Every country establishes regulations for maximum acceptable concentrations as well as desiderated ones. Pollutants of major public health concern include nitrogen dioxide, sulfur dioxide, and ammonia.

Ammonia (NH_3_) is a pungent-smelling and toxic gas naturally present in the environment, with a typical concentration of a few parts per billion (ppb). Its excessive release into the atmosphere through human activities has resulted in it being considered a significant air pollutant. Its concentrations should be controlled below 20 parts per million (ppm), otherwise it can lead to an irritating effect on the eyes and on the respiratory system, which becomes critical at a concentration of 110 ppm [5]. Above 1500 ppm, NH_3_ is very harmful or even lethal to humans [6].

Nitrogen dioxide (NO_2_) is a reddish-brown and pungent-smelling gas, primarily produced by the combustion of fossil fuels, including combustion in vehicle engines and industrial processes. It reacts with water to produce nitric acid and nitric oxide as well as with other pollutants in the atmosphere to form fine particulate matter (PM2.5). Different limit values of NO_2_ have applied to outdoor air. For a one hour exposure period, the European Union has specified a value of 100 ppb as the maximum concentration limit [7]. Inhalation of NO₂ can cause dizziness, headaches, and irritation of the respiratory system. Prolonged exposure to elevated levels of NO_2_ can exacerbate respiratory conditions such as asthma and shortness of breath.

Sulfur dioxide (SO_2_) is a colorless, pungent-smelling, and irritant gas mainly produced in fossil fuel combustion and industrial processes. SO_2_ is responsible for acid rain and can contribute to the formation of PM2.5. SO_2_ occurs in concentrations typically of up to several tens of ppb and in extreme cases up to several hundreds of ppb [8]. The exposure to SO_2_ can have serious health implications, causing acute severe effects in the bronchi at concentrations higher than 4 ppm, and prolonged exposure can compromise lung function [9].

Real-time pollution monitoring with high sensitivity and selectivity as well as public alerts is vital to minimize the exposure of the population, particularly the vulnerable categories, to air pollution [10,11]. Miniature, low-cost electrochemical sensors have found widespread use in industrial settings but generally are still not stable or sensitive enough for monitoring ambient pollutants reliably. Photonics sensors based on laser absorption spectroscopy are of growing interest due to the development of powerful infrared lasers, such as quantum cascade lasers (QCLs) tunable over wide wavelength ranges, which permit the unambiguous detection of numerous substances at low concentrations [12,13]. Several optical-based sensors have been developed to detect air pollutants, with the final aim of on-field, real-time detection. Trace concentrations of NH_3_, NO_2_, and SO_2_ were measured by employing detection techniques based on direct detection of absorbed light, such as tunable diode laser absorption spectroscopy [14,15,16] and cavity ringdown spectroscopy [17,18,19]. Among the optical techniques for gas sensing, Quartz-Enhanced Photoacoustic Spectroscopy (QEPAS) exploits the photoacoustic effect occurring in a gas sample when a modulated, resonant light is absorbed by the target analytes. A weakly damped propagating acoustic (pressure) wave with wavelengths in the centimeter range is generated in the proximity of the exciting light beam [20]. In QEPAS, these sound waves are detected by a spectrophone, composed of a quartz-tuning fork (QTF) transducer and a pair of millimeter-size resonator tubes, aligned on both sides of the QTF. The laser beam is focused between the QTF prongs while passing through both tubes. In this way, the sound wave is generated between QTF’s prongs and confined in the resonator tubes. The generated standing wave vibrational pattern puts prongs in vibration and a sinusoidal electrical signal is generated because of the piezoelectric effect occurring in the quartz crystal [21]. Avoiding the use of a photodetector, QEPAS is a wavelength-independent technique. Indeed, the same QTF can operate with laser sources emitting in the spectral range from UV to THz. This classifies QEPAS as an ideal technique for multi-gas detection when it is combined with a multi-laser source [22,23,24,25]. 

In this work, a multi-QCL QEPAS sensor for multi-gas detection was developed and tested in a laboratory environment. A custom-designed three-wavelength laser module was employed as a light source for the multi-color QEPAS sensor. This module consists of three different QCLs whose output beams are combined into a single collimated output beam by means of dichroitic beam combiners mounted inside the module. In this configuration, the developed QEPAS sensor was calibrated and tested for sequential detection of gaseous samples containing trace concentrations of NO_2_, SO_2_, and NH_3_.

## 2. Assembly of the Multi-QCL Source

The three-wavelength laser module integrates three distinct QCL chips and is designed to emit a single collimated beam output by using dichroitic beam combiners. The three Distributed Feedback QCLs from Thorlabs GmbH (Newton, MA, USA) assembled in the module are the QD7385HHLH-B device designed for operation at 7.38 μm, the QD9062HHLH-C operating at 9.06 μm, and the QD6250HHLH-A emitting at 6.25 μm. The three QCLs were selected to target absorption features of SO_2_, NH_3_, and NO_2_, respectively.

A schematic illustration of the 3λ-QCL module prototype is depicted in Figure 1a.

F1 is a low-pass filter (Spectrogon SP-7500, Täby, Sweden) with a transmission spectrum that allows the transmission of the 6.25 μm beam and the reflection of the 9.06 μm beam. On the other hand, F2 is a band-pass filter (Spectrogon BBP-7000-8750c, Täby, Sweden) used for the transmission of the 7.38 μm beam and efficient reflection the other two beams. For this reason, F2 is placed in front of the 7.38 μm QCL. F1, F2, and the mirror M1 are attached to kinematic mounts (IM10.C2 from Siskiyou Corporation, Redwood, CA, USA) with two adjusters with an angular range of ±4°. Moreover, the F1 and M1 kinematic mounts are fixed on a 1-axis translation stage (50.25 dt from Siskiyou Corporation) with a 6 mm adjustment range to fine control the distance between the filter and the mirror with the QCLs. Thus, fine tilting of M1, F1, and F2 can steer both the 9.06 μm and the 6.25 μm beam to make them collinear with the 7.38 μm beam. In this way, the three QCL beams can come out of the output module window collinearly. The three QCL sources are mounted on three custom-made heatsinks for extra heating dissipation when the lasers are turned on. The module is mounted over a back plate with four holes for easy fitting with an optical table. The lateral sides can be easily removed to ensure quick access to the internal components for fine adjustments of the optical alignment. A 3D-sketch of the module is represented in Figure 1b. 

The alignment of the three QCL beams plays a crucial role in their use as a light source in a QEPAS sensor. It is mandatory to reach high collinearity of the three QCL beams in order to use a single aspheric lens to simultaneously focus all of them in an 800 µm-wide gap between the prongs of the QTF without touching the QTF prongs. The exact alignment and focus allows for the noise contribution coming from light touching the QTF prong to be minimized and for increased accuracy and precision of measurements. 

A pyrocamera (Pyrocam III, Ophir Spiricon, Darmstadt, Germany) with high-resolution pixel dimensions of 0.08 × 0.08 mm was used for the beam alignment and for acquiring and analyzing far field beam profiles at different distances from the 3λ-QCL module. The collinearity of the three beams was firstly evaluated by analyzing their spatial overlap at 25 cm from the 3λ-QCL module, where the focusing lens of the QEPAS sensor was supposed to be located. The beam profiles were acquired in pairs, namely, the 7.38 μm beam together with the 9.06 μm beam and the 7.38 μm beam with the 6.25 μm beam, as shown in Figure 2a,b.

The results show a very good overlap of the profiles at a 25 cm distance from the 3λ-QCL module. Then, a 1′′-diameter ZnSe Plano-Convex lens with a focal length of 50 mm and an AR coating in the range of 7–12 μm (model LA7656-E3 from Thorlabs; the same lens was mounted in the QEPAS sensor) was placed at a 25 cm distance from the 3λ-QCL module, with the pyrocamera moved into the focal plane of the lens. To investigate the characteristics of the laser beams in the focal plane, beam spots were acquired in pairs, as well as for the measurements reported in Figure 2a,b. The results are shown in Figure 3a,b. The beam waists were measured by extracting the intensity values along the x- and y-directions of the 2D-intensitiy distribution. For both directions, the extracted datapoints were fitted with a Gaussian function. For each gaussian fit, the beam waist was evaluated as the radial distance at which the light intensity dropped to 1/e^2^ of its maximum central value. The extracted beam waist values were 350 μm, 430 μm, and 310 μm for the 7.38 μm, 9.06 μm, and 6.25 μm beam spots, respectively. For ease of viewing, each spot is sketched in Figure 3c as a circumference, with the mean value of beam waists along the x- and y-directions as the diameter and the coordinates of the peak value those of the center. The data in Figure 3c clearly show that the distance between the three peak values of the three beams was less than 50 µm, comparable to the pyrocamera resolution. The excellent degree of overlap as well as the similarity of the beam spot sizes themselves allowed for easy alignment of the three beams through the spectrophone of the QEPAS sensor, as discussed in the next session.

## 3. Architecture of the 3λ-QEPAS Sensor

The 3λ-QEPAS sensor architecture is depicted in Figure 4.

The 3λ-QCL module was used as a light source exciting the analytes within the acoustic detection module (ADM01, provided by Thorlabs GmbH, Newton, USA). It is composed of the QEPAS spectrophone enclosed in stainless-steel housing with an inlet and outlet connector for gas flow. The QEPAS spectrophone consists of a T-shaped QTF and a pair of resonator tubes aligned in an on-beam configuration. Each tube has a length of 12.4 mm and an internal diameter of 1.59 mm. The laser beam was focused into the ADM and fixed on a five-axis stage for alignment purposes using a 50 mm focal-length ZnSe lens with a 3–12 µm AR coating. QEPAS measurements were performed using the wavelength modulation and dual-frequency detection method (2f-WM): A sinusoidal dither matching half of the QTF resonance frequency of the employed spectrophone was applied to the QCL current driver (ITC4002QCL, Benchtop Laser Driver and Temperature Controller, Thorlabs) and the transduced QTF signal was demodulated by the lock-in amplifier (MFI 500 kHz Lock-in Amplifier, Zurich Instruments, Zurich, Switzerland) at the QTF resonance frequency [26].

The lock-in integration time was set to 100 ms. The demodulated signal was thus digitized and stored on a personal computer by means of a data acquisition board, with the card-sampling time set to three times the lock-in integration time. The pressure of the gas mixtures flowing inside the ADM was regulated using a pressure controller, a valves system, and a vacuum pump, whereas the flow rate was set by the gas mixer (MCQ Instruments, Gas Blender 103, Rome Italy), with a setpoint accuracy of 1% for each channel. All measurements were performed by fixing the mass flow rate to 90 sccm and the pressure to 400 Torr, resulting from preliminary measurements of the optimal operating pressure for the sequence detection of the three gas species, namely, NH_3_, SO_2_, and NO_2_, of 9.06 μm, 7.38 μm, and 6.25 μm QCL, respectively. At 400 Torr, the spectrophone had a resonance frequency of 12,439.4 Hz with a quality factor of 14,650.

## 4. Results

### 4.1. Single-Analyte Calibration

The 3λ-QCL QEPAS sensor was calibrated to detect the selected absorption species, i.e., NH_3_, NO_2_, and SO_2_. First, the sensor was operated to target each absorber independently. The optimal absorption lines were selected to provide both the highest intensity and the lowest interference with spectral lines of other absorbers. 

For each of three analytes, the absorption cross-section was reconstructed within the spectral dynamic range of the related QCL by using the HITRAN database to simulate a mixture of 10 ppm of the analyte in N_2_ [27]. To ensure that the other two analytes, H_2_O, and other standard air components did not interfere with the detection of the selected analyte, mixtures consisting of 10 ppm of the other two analytes in N_2_ and a mixture of 1% of H_2_O in standard air were also simulated within the same spectral range of the QCL. The results of the HITRAN simulations are reported in Figure 5a–c for each target analyte.

For each analyte, the 3λ-QCL QEPAS sensor calibration was performed in controlled humidity conditions by setting the water vapor concentration to 1.5 ‰. The humidity level and the temperature of the gas mixture within the gas line were continuously monitored by a hygrometer, i.e., iST humidity module HYT 271(Innovative Sensor technology, Ebnat-Kappel, Switzerland). The analyte concentration inside the humidified line was varied by diluting a certified concentration of the analyte in N_2_ with pure N_2_. When a mixture with a fixed analyte concentration is injected into the gas line for sensor calibration, polar molecules with a permanent dipole moment have a strong tendency to stick to surfaces with adsorption and/or desorption processes, leading to a time-dependent gas-phase concentration within the gas line. For example, an extensive study of the effect of the stickiness of NH_3_ molecules on the estimation of their concentration is reported in Ref. [23]. The study demonstrated that, starting from a fixed concentration injected into the gas line, a dynamic equilibrium between the gas flux and the adsorption processes could be established after a transient, leading to a stable concentration flowing into the ADM. A similar situation could be established in the sensor-cleaning process. For the three gas species, the equilibrium condition was reached in less than 20 min, mainly due to the high flow rate employed (90 sccm). For each gas species, before passing to the next concentration, the gas line was purged with pure nitrogen for about 1 h. For each analyte concentration value, a spectral scan around the peak was performed with a 100 ms lock-in integration time. The measured 2f-QEPAS spectral scans are shown in Figure 6a–c. 

For each spectral scan, peak values of the strongest feature (occurring at 392 mA, 298 mA, and 257 mA for NH_3,_ NO_2_, and SO_2_, respectively) were extracted and plotted as a function of the concentration for each analyte to obtain the calibration curves reported in Figure 7a–c with the best linear fits (red solid lines) of the experimental data. A strong correlation between the data and the best linear fit was derived for all the three calibration curves, with a coefficient of determination of R^2^ > 0.999. Sensitivities (slopes of the linear fits) of 19.99 ± 0.30 mV/ppm, 19.39 ± 0.19 mV/ppm, and 73.99 ± 0.49 mV/ppm were estimated for NO_2_, SO_2_, and NH_3_ detection, respectively, with a measured 1-σ noise level of 0.18 mV. The best linear fit of the NO_2_ QEPAS sensor calibration (Figure 7b) indicated a negative intercept (−15.13 ± 1.85 mV) far from the zero value. This is ascribed to the chemical reaction of the NO_2_ molecules with the H_2_O molecules within the gas line, which resulted in an effective decrease in the actual concentration of NO_2_ molecules reaching the ADM [28].

This phenomenon showed up as a shift of the calibration curve along the *x*-axis, causing the negative intercept. An accurate estimation of the NO_2_ concentration considering this behavior required a concentration correction of 760 ppb. The minimum detection limit (MDL) was calculated as the concentration corresponding to a signal-to-noise ratio equal to 1. Therefore, MDLs of 9 ppb, 9.3 ppb, and 2.4 ppb was estimated for NO_2_, SO_2_, and NH_3_ detection, respectively, with a lock-in integration time of 100 ms.

To investigate the stability of the developed sensor, an Allan–Werle deviation analysis of the QEPAS signal was performed [29]. Each laser source was operated at a fixed current far from the absorption features, with a 2 h-long acquisition of QEPAS signals in pure N_2_ performed at a 0.1 s lock-in integration time. The measurements were taken under the same experimental condition set for the sensor’s calibration. The results are shown in Figure 8.

When each QCL was turned on, the noise level decreased as the integration time increased up to 100 s, following the expected trend of $\sim 1/\sqrt{t}$, which indicates that the main contribution to the sensor’s noise was due to the QTF thermal noise. For integration times higher than 100 s, the noise started to deteriorate. This behavior is ascribed to mechanical instabilities of the sensor, i.e., laser instability, mechanical vibrations, etc., which affect the performance of the sensor during long integration times. Indeed, the turning point at 100 ms was not observed when the three QCLs were simultaneously off, strengthening the hypothesis that the turning point is due to sensor instabilities induced by operation of the laser. At an average time of 10 s, MDLs of 0.4 ppb, 1.5 ppb, and 1.4 ppb were achieved for NH_3_, NO_2_, and SO_2_, respectively. 

### 4.2. Multi-Gas Detection

The 3λ-QCL multi-gas detection employing the 3λ-QCL QEPAS sensor was performed by flowing three different gas mixtures in the ADM:Mix #1: 5 ppm NH_3_, 5 ppm NO_2_, 2.1 ppm SO_2_ in N_2_;Mix #2: 10 ppm NH_3_, 2.5 ppm NO_2_, 2.1 ppm SO_2_ in N_2_;Mix #3: 5 ppm NH_3_, 2.5 ppm NO_2_, 4.3 ppm SO_2_ in N_2_.

For each gas mixture, the spectral scan of each gas species was acquired with a 100 ms lock-in integration time under the same experimental conditions described in the previous sections. The measured 2f-QEPAS spectral scans are shown in Figure 9a–i.

The spectral scans acquired in the multi-gas mixtures retraced those acquired in the single-gas mixtures (see Figure 6a–c). No relevant variations in the spectra due to the presence of the other components in the samples were observed. The calibration curves extracted in Figure 7a–c were employed to estimate the target gas concentrations in the three mixtures. The peak values for each species were extracted, and the QEPAS signals were converted into gas concentrations using the sensitivity values, for each analyte. The achieved results are reported in Table 1 together with the expected concentrations for each analyte.

The errors associated with the estimated concentrations were calculated by propagating the uncertainties associated with the retrieved sensitivities and with the collected signals. The latter were evaluated as 1-σ standard deviation of long-term acquisition (~30 min) performed by flushing the multi-gas mixtures in the QEPAS sensor (relative fluctuations below 1% of the corresponding signal). The errors associated with the expected concentrations were calculated by considering the uncertainties of certified concentrations for the gas cylinders with relative uncertainties of 4% at 3-σ and the uncertainties provided by the gas mixer. The collected results clearly indicate that the retrieved concentrations were in excellent agreement with the expected ones. For each gas sample, the relative deviation of the predicted concentrations was calculated, resulting in an average relative deviation of 96.6%, 98.6%, and 99.9% for Mix #1, #2, and #3, respectively. 

## 5. Conclusions

In this work, a multi-QCL QEPAS sensor employing a three-wavelength laser module was successfully implemented and experimentally validated in a controlled laboratory environment. The module incorporates three Distributed Feedback QCLs supplied by Thorlabs GmbH, each operating at distinct wavelengths of 7.38 μm, 9.06 μm, and 6.25 μm. The 3λ-QCL module was employed as a light source in a QEPAS sensor for the detection of NH_3_, NO_2_, and SO_2_ in sequence. Sensitivities of 73.99 mV/ppm, 19.99 mV/ppm, and 19.39 mV/ppm were measured for NH_3_, NO_2_, and SO_2_, respectively, with a measured 1-σ noise level of 0.18 mV. Therefore, MDLs of 2.4 ppb, 9 ppb, and 9.3 ppb were estimated, respectively, which were well below their typical natural abundance in air, even after demodulating the collected signal at an integration time as low as 0.1 s. 

Multi-gas detection employing the 3λ-QCL QEPAS sensor was also performed by targeting three different gas mixtures, retracing the same experimental conditions employed for the single-analyte mixtures. The presence of other components in the gas samples did not introduce any substantial variations in the spectra. The developed 3λ-QCL QEPAS sensor demonstrated its capability to accurately determine the composition of the gas samples, with an average accuracy of >96.6%. 

The compact multi-QCL assembly, as well as the compactness of the QEPAS sensors, promotes the development of a multi-species QEPAS sensor for accurate, precise, and reliable real-time outdoor air pollutant monitoring.

## Figures and Tables

**Figure 1 sensors-23-09005-f001:**
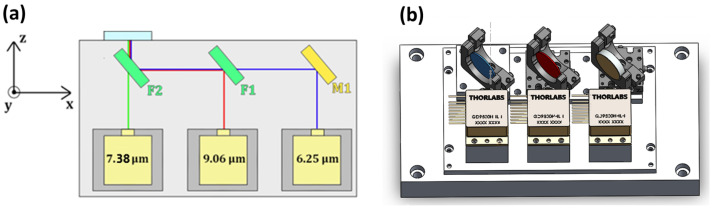
(**a**) Schematic of the internal structure of the 3λ-QCL module. F2 is a band-pass filter; F1 is a low-pass filter; M1 is a mirror. (**b**) Top view of the Solidworks 3D model of the 3λ-QCL module without the top and lateral sides.

**Figure 2 sensors-23-09005-f002:**
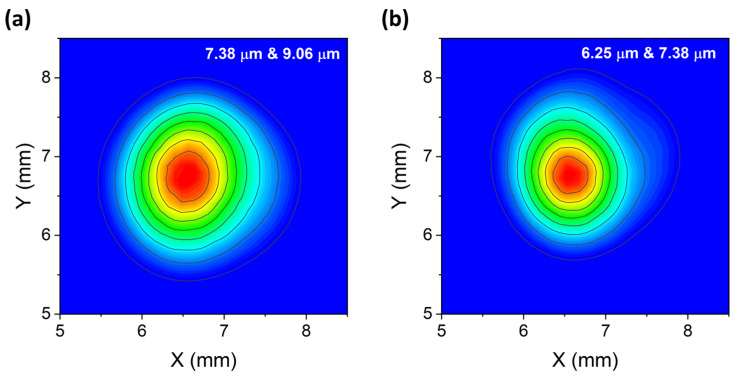
(**a**) Combined 7.38 μm and 9.06 μm beam profiles overlapped at 25 cm from the 3λ-QCL module. (**b**) Combined 7.38 μm and 6.25 μm beam profiles overlapped at 25 cm from the 3λ-QCL module.

**Figure 3 sensors-23-09005-f003:**
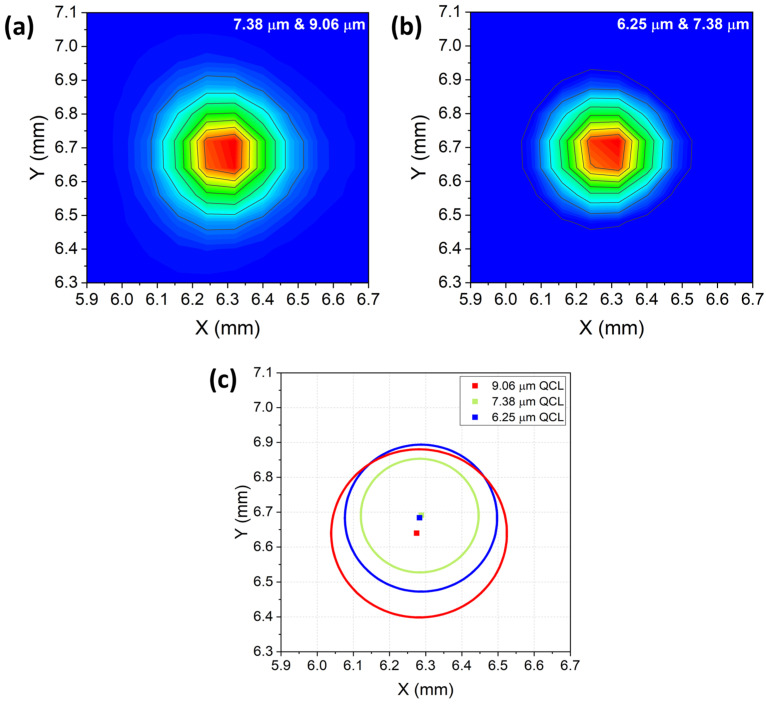
(**a**) Combined 7.38 μm and 9.06 μm beam profiles overlapped at the focal plane of the lens. (**b**) Combined 7.38 μm and 6.25 μm beam profiles overlapped at the focal plane of the lens. (**c**) Representation of the three beam spots as circumferences. The 7.41 µm, 6.25 µm, and 9.06 µm beam waists are depicted as green, blue, and red solid circumferences, respectively. The radii of the circumferences are equal to the mean value of widths of the beam waists along the x- and y-directions. The coordinates of the center of the circumferences are those of the peak values of the three beam spots.

**Figure 4 sensors-23-09005-f004:**
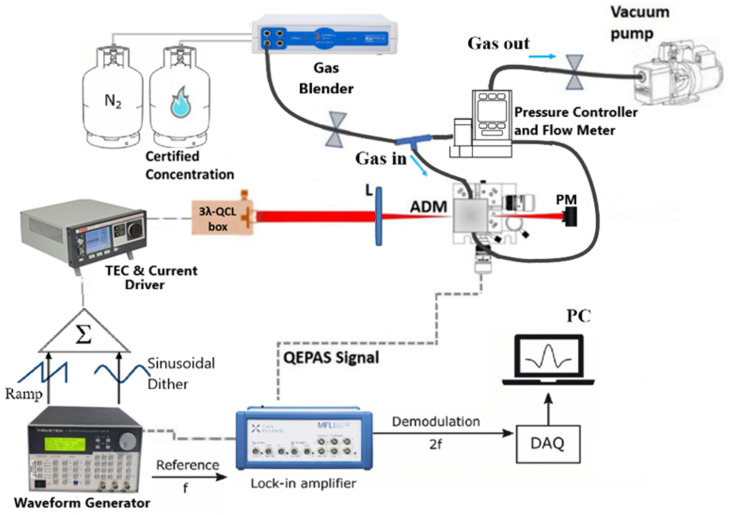
Schematic of the 3λ-QCL-based QEPAS sensor for NH_3_, SO_2_, and NO_2_ detection. L—lens; ADM—acoustic detection module; DAQ—data acquisition board; PC—personal computer; PM—power meter.

**Figure 5 sensors-23-09005-f005:**
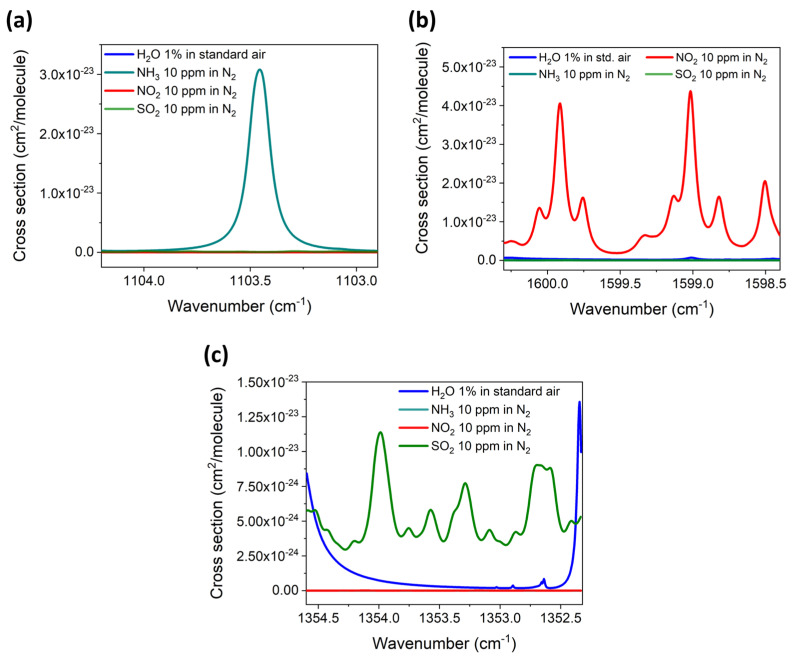
(**a**) HITRAN simulation of an absorption cross-section of a mixture of 10 ppm of NH_3_ in N_2_, a mixture of 10 ppm of SO_2_ in N_2_, a mixture of 10 ppm of NO_2_ in N_2_, and a mixture of 1% of water vapor in standard air within the emission spectral range of the 9.06 μm QCL. (**b**) Simulation of an absorption cross-section of a mixture of 10 ppm of NO_2_ in N_2_, a mixture of 10 ppm of SO_2_ in N_2_, a mixture of 10 ppm of NH_3_ in N_2_, and a mixture of 1% of water vapor in N_2_ within the emission spectral range of the 6.25 μm QCL. (**c**) Simulation of an absorption cross-section of a mixture of 10 ppm of SO_2_ in N_2_, a mixture of 10 ppm of NH_3_ in N_2_, a mixture of 10 ppm of NO_2_ in N_2_, and a mixture of 1% of water vapor in N_2_ within the emission spectral range of the 7.38 μm QCL.

**Figure 6 sensors-23-09005-f006:**
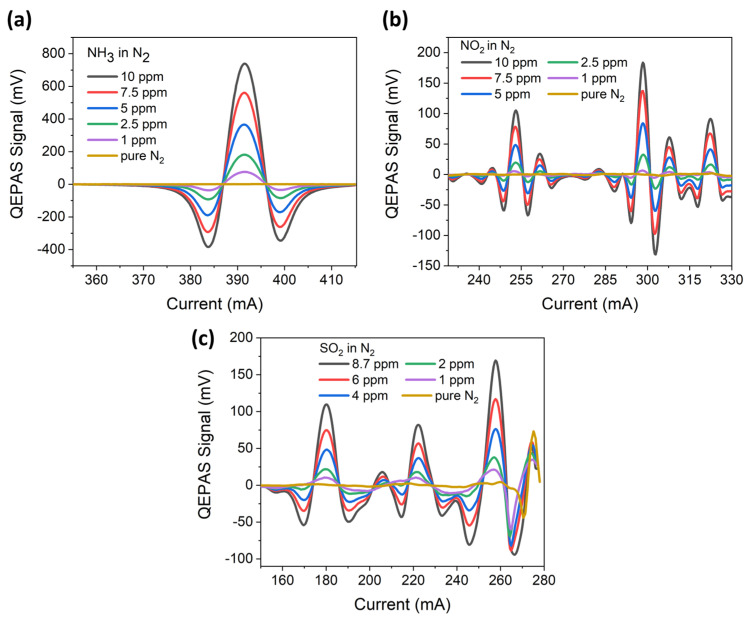
(**a**) QEPAS spectral scans measured for different concentrations of NH_3_ in N_2_ and pure N_2_ using the 9.06 μm QCL. (**b**) QEPAS spectral scans measured for different concentrations of NO_2_ in N_2_ and pure N_2_ obtained when the 6.25 μm QCL is turned on. (**c**) QEPAS spectral scans measured for different concentrations of SO_2_ in N_2_ and pure N_2_ using the 7.38 μm QCL The peak at 275 mA observed for pure N_2_ is due to residual H_2_O in the gas line.

**Figure 7 sensors-23-09005-f007:**
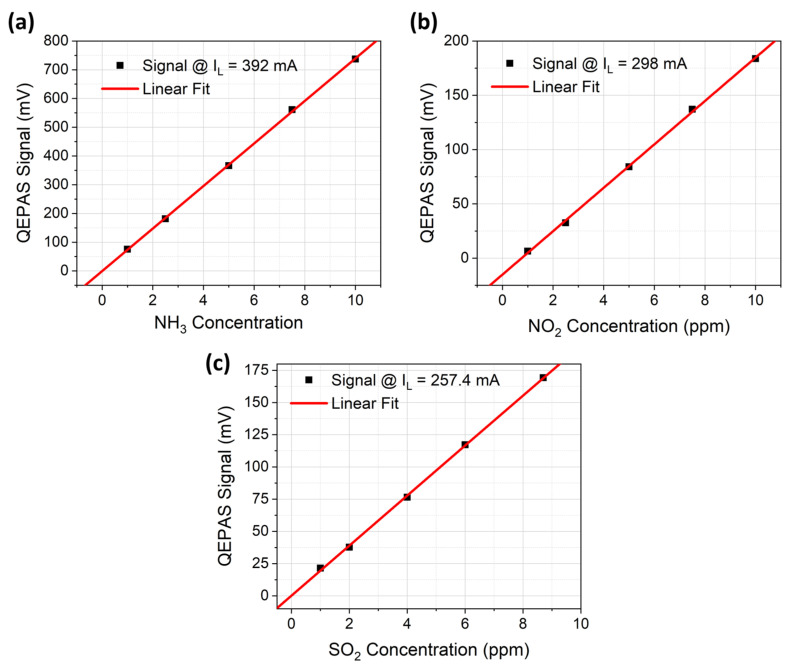
(**a**) QEPAS signal as a function of the NH_3_ concentration (black squares) with the corresponding best linear fit (red line). (**b**) QEPAS signal as a function of the NO_2_ concentration (black squares) with the corresponding best linear fit (red line). (**c**) QEPAS signal as a function of the SO_2_ concentration (black squares) with the corresponding best linear fit (red line).

**Figure 8 sensors-23-09005-f008:**
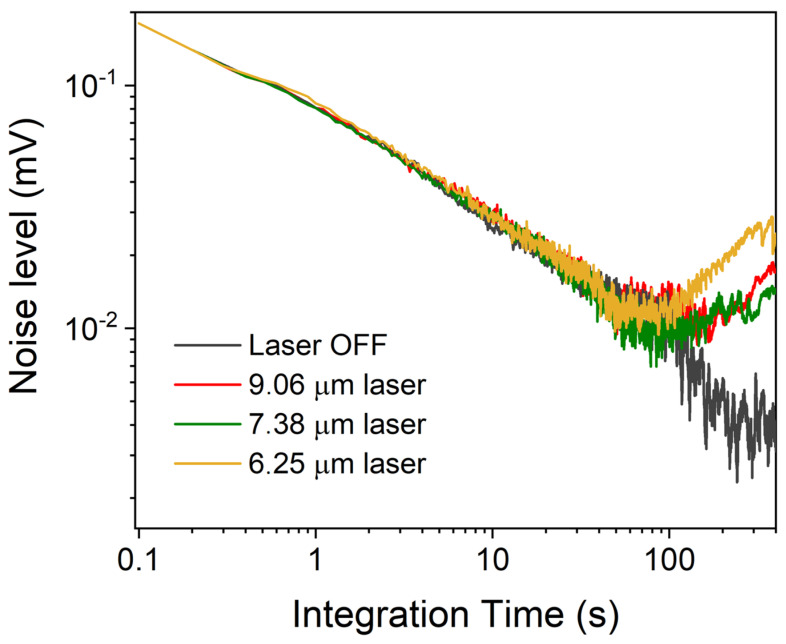
Allan deviation of the QEPAS signal as a function of the lock-in integration time.

**Figure 9 sensors-23-09005-f009:**
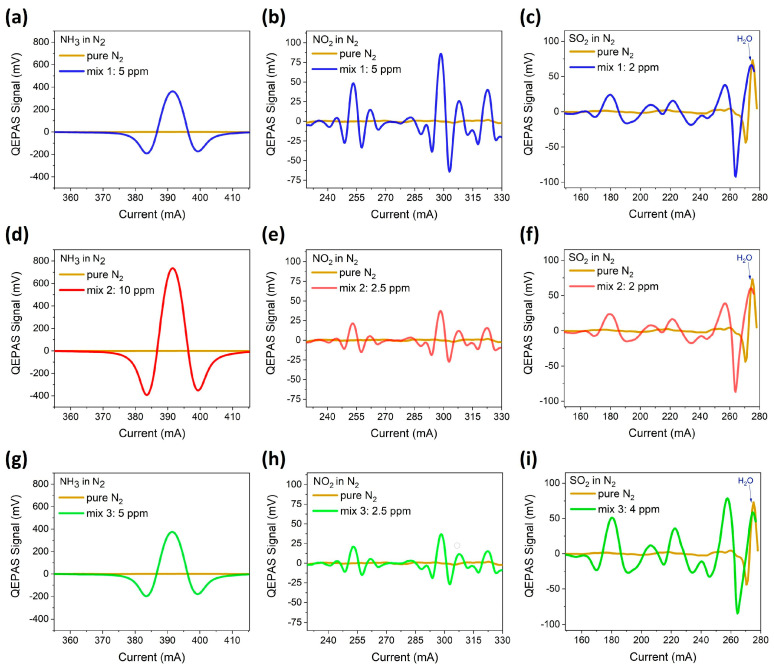
QEPAS spectral scan of NH_3_ (**a**), NO_2_ (**b**), and SO_2_ (**c**) in Mix #1 NO_2_; NH_3_ (**d**), NO_2_ (**e**), and SO_2_ (**f**) in Mix #2; NH_3_ (**g**), NO_2_ (**h**), and SO_2_ (**i**) in Mix #3. The peak at 275 mA observed for pure N_2_ is due to residual H_2_O in the gas line.

**Table 1 sensors-23-09005-t001:** Expected and estimated analyte concentrations in the three analyzed mixtures.

Mix	Target Gas	Expected Concentration (ppm)	Estimated Concentration (ppm)
#1	NH_3_	5.0 ± 0.1	4.90 ± 0.03
#1	NO_2_	5.0 ± 0.1	5.06 ± 0.08
#1	SO_2_	2.1 ± 0.1	1.97 ± 0.03
#2	NH_3_	10.0 ± 0.2	9.96 ± 0.07
#2	NO_2_	2.5 ± 0.1	2.62 ± 0.05
#2	SO_2_	2.1 ± 0.1	2.01 ± 0.02
#3	NH_3_	5.0 ± 0.1	5.06 ± 0.04
#3	NO_2_	2.5 ± 0.1	2.60 ± 0.05
#3	SO_2_	4.3 ± 0.1	4.06 ± 0.05

## Data Availability

Row data will be available on demand.

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
