# Peer review of "Quartz-Enhanced Photoacoustic Sensor Based on a Multi-Laser Source for In-Sequence Detection of NO2, SO2, and NH3"

_sensors, 2023, doi:10.3390/s23219005_

Round 1

Reviewer 1 Report

Comments and Suggestions for Authors

1) The title needs to be modified into a statement that readers can understand at a glance. I suggest changing it to "Enhanced Multi-Laser Quartz Photoacoustic Sensor for Simultaneous Detection of NO2, SO2, and NH3 Air Pollutants".

2) Starting from line 58, this paragraph introduces the existing monitoring methods for NO2, SO2, and NH3. The description of the existing work is not sufficient. I suggest expanding it to cover ground-based measurement instruments to satellite-based monitoring of these three pollutants, such as Sentinel-5, and so on.

3) Starting from line 81, the author's work motivation and innovation are not clear enough. I suggest further clarification.

4) In the second section, the instruments with three wavelengths need to be clearly defined. Specify which wavelengths are used and explain why they are referred to as three wavelengths.

5) Figure 2, what do the two red solid lines represent? Are they a comparison of particle diameter and resolution?

6) In the third section, were these instruments built by your team, or are they products of an existing company? If they were built for experimentation purposes, I suggest changing the title to "Experimental Setup Construction".

7) There are too few sample points in Figure 7. The fitting is not convincing with such a small number of measurements. I suggest adding several thousand more measurement points before fitting.

8) In Figure 8, why does the noise level increase again after an integration time of over 100 seconds?

Comments on the Quality of English Language

 Moderate editing of English language required

Author Response

Our comments to reviewer's report are here attached.

Reviewer 2 Report

Comments and Suggestions for Authors

The manuscript (sensors-2654074) examines the application of a multiquantum cascade laser (QCL) module in quartz-enhanced photoacoustic spectroscopy (QEPAS) for gas detection. The module, comprising three distinct QCLs, is united with a dichroitic beam combiner to create an overlapping, collimated beam. In a laboratory setting, the system was evaluated for its ability to detect NO2, SO2, and NH3, attaining sensitivities of 19.99 mV/ppm, 19.39 mV/ppm, and 73.99 mV/ppm, respectively. With an integration time of 100 ms, the detection limits for these gases were 9 ppb, 9.3 ppb, and 2.4 ppb, respectively—figures below the typical natural abundance of these gases in the atmosphere.

The manuscript is very well-written, with informative, clear, and suitable sections for publication. The authors demonstrate the analyses in a conceptual, technical, and well-articulated manner. The presented images and tables are clear and do not require corrections. The only recommendation is to check old references.

Comments on the Quality of English Language

Minor changes in grammar are needed.

Author Response

We thank the reviewer for its comments. According to its suggestions we added some more recent references in the manuscript and performed a grammar check.

Reviewer 3 Report

Comments and Suggestions for Authors

This manuscript presented a QEPAS sensor coupling with three lasers for NO2, SO2 and NH3 measurement, and the detection limits reached down to ppb levels. The manuscript is well written and has a good structure. But some important points (see below) need to be clarified and the manuscript needs to be modified below publication.

1, Gas temperature measurement in the ADM is missing, how to ger the gas temperature is not mentioned. Without knowing the temperature for QEPAS sensor, it will be a problem. The author needs to explain.

2, For NH3 and NO2, especially for NH3, there will be molecule losses to the inner surface of the system due to the sticky effect of NH3. The author needs to provide more detailed information on e.g. how to passive the system (how many time, for how long) to reach the equilibrium condition.

3, Fig.7, the slope of the linear fit is used for calibration, what is the uncertainty of the slope (including both x-error and y-error bars)? This uncertainty will have direct effect on the uncertainty of your final results. This number should be given here.

4, what is the uncertainty of the gas mixer, this is missing.

5, Table 1, the uncertainty of estimated concentrations were given, but how these numbers were calculated? This should be included.   

Comments on the Quality of English Language

Good

Author Response

(The authors gave the same response as above.)

Round 2

Reviewer 1 Report

Comments and Suggestions for Authors

The authors answered the comments and clarified some issues.

Comments on the Quality of English Language

Minor editing of the English language is needed to make some content more clear.

Reviewer 3 Report

Comments and Suggestions for Authors

The authors have addressed all the comments, and revised the manuscript.